# THE POWER OF DEEPER NETWORKS FOR EXPRESSING NATURAL FUNCTIONS

**David Rolnick, Max Tegmark**
Massachusetts Institute of Technology
`{drolnick, tegmark}@mit.edu`

## ABSTRACT

It is well-known that neural networks are universal approximators, but that deeper networks tend in practice to be more powerful than shallower ones. We shed light on this by proving that the total number of neurons $m$ required to approximate natural classes of multivariate polynomials of $n$ variables grows only linearly with $n$ for deep neural networks, but grows exponentially when merely a single hidden layer is allowed. We also provide evidence that when the number of hidden layers is increased from 1 to $k$, the neuron requirement grows exponentially not with $n$ but with $n^{1/k}$, suggesting that the minimum number of layers required for practical expressibility grows only logarithmically with $n$.

## 1 INTRODUCTION

Deep learning has lately been shown to be a very powerful tool for a wide range of problems, from image segmentation to machine translation. Despite its success, many of the techniques developed by practitioners of artificial neural networks (ANNs) are heuristics without theoretical guarantees. Perhaps most notably, the power of feedforward networks with many layers (*deep* networks) has not been fully explained. The goal of this paper is to shed more light on this question and to suggest heuristics for how deep is deep enough.

It is well-known (Cybenko, 1989; Funahashi, 1989; Hornik et al., 1989; Barron, 1994; Pinkus, 1999) that neural networks with a single hidden layer can approximate any function under reasonable assumptions, but it is possible that the networks required will be extremely large. Recent authors have shown that some functions can be approximated by deeper networks much more efficiently (i.e. with fewer neurons) than by shallower ones. Often, these results admit one or more of the following limitations: "existence proofs" without explicit constructions of the functions in question; explicit constructions, but relatively complicated functions; or applicability only to types of network rarely used in practice.

It is important and timely to extend this work to make it more concrete and actionable, by deriving resource requirements for approximating natural classes of functions using today's most common neural network architectures. Lin et al. (2017) recently proved that it is exponentially more efficient to use a deep network than a shallow network when Taylor-approximating the product of input variables. In the present paper, we move far beyond this result in the following ways: (i) we use standard uniform approximation instead of Taylor approximation, (ii) we show that the exponential advantage of depth extends to all general sparse multivariate polynomials, and (iii) we address the question of how the number of neurons scales with the number of layers. Our results apply to standard feedforward neural networks and are borne out by empirical tests.

Our primary contributions are as follows:

- It is possible to achieve **arbitrarily close approximations** of simple multivariate and univariate polynomials with neural networks having a **bounded number of neurons** (see §3).

- Such polynomials are **exponentially easier to approximate with deep networks than with shallow networks** (see §4).

- The power of networks improves rapidly with depth; for natural polynomials, **the number of layers required is at most logarithmic** in the number of input variables, where the base of the logarithm depends upon the layer width (see §5).

## 2 RELATED WORK

Deeper networks have been shown to have greater representational power with respect to various notions of complexity, including piecewise linear decision boundaries (Montufar et al., 2014) and topological invariants (Bianchini & Scarselli, 2014). Recently, Poole et al. (2016) and Raghu et al. (2016) showed that the trajectories of input variables attain exponentially greater length and curvature with greater network depth.

Work including Daniely (2017); Eldan & Shamir (2016); Pinkus (1999); Poggio et al. (2017); Telgarsky (2016) shows that there exist functions that require exponential width to be approximated by a shallow network. Barron (1994) provides bounds on the error in approximating general functions by shallow networks. Mhaskar et al. (2016) and Poggio et al. (2017) show that for *compositional functions* (those that can be expressed by recursive function composition), the number of neurons required for approximation by a deep network is exponentially smaller than the best known upper bounds for a shallow network. Mhaskar et al. (2016) ask whether functions with tight lower bounds must be pathologically complicated, a question which we answer here in the negative.

Various authors have also considered the power of deeper networks of types other than the standard feedforward model. The problem has also been posed for sum-product networks (Delalleau & Bengio, 2011) and restricted Boltzmann machines (Martens et al., 2013). Cohen et al. (2016) showed, using tools from tensor decomposition, that shallow arithmetic circuits can express only a measure-zero set of the functions expressible by deep circuits. A weak generalization of this result to convolutional neural networks was shown in Cohen & Shashua (2016).

## 3 THE POWER OF APPROXIMATION

In this paper, we will consider the standard model of feedforward neural networks (also called *multilayer perceptrons*). Formally, the network may be considered as a multivariate function $N(\mathbf{x}) = \mathbf{A}_k \sigma(\cdots \sigma(\mathbf{A}_1 \sigma(\mathbf{A}_0 \mathbf{x})) \cdots)$, where $\mathbf{A}_0, \mathbf{A}_1, \ldots, \mathbf{A}_k$ are constant matrices and $\sigma$ denotes a scalar nonlinear function applied element-wise to vectors. The constant $k$ is referred to as the *depth* of the network. The *neurons* of the network are the entries of the vectors $\sigma(\mathbf{A}_\ell \cdots \sigma(\mathbf{A}_1 \sigma(\mathbf{A}_0 \mathbf{x})) \cdots)$, for $\ell = 1, \ldots, k-1$. These vectors are referred to as the *hidden layers* of the network.

Two notions of approximation will be relevant in our results: $\epsilon$-*approximation*, also known as *uniform approximation*, and *Taylor approximation*.

**Definition 3.1.** *For constant $\epsilon > 0$, we say that a network $N(\mathbf{x})$ $\epsilon$-approximates a multivariate function $f(\mathbf{x})$ (for $\mathbf{x}$ in a specified domain $(-R, R)^n$) if $\sup_{\mathbf{x}} |N(\mathbf{x}) - f(\mathbf{x})| < \epsilon$.*

**Definition 3.2.** *We say that a network $N(\mathbf{x})$ Taylor-approximates a multivariate polynomial $p(\mathbf{x})$ of degree $d$ if $p(\mathbf{x})$ is the $d$th order Taylor polynomial (about the origin) of $N(\mathbf{x})$.*

The following proposition shows that Taylor approximation implies $\epsilon$-approximation for homogeneous polynomials. The reverse implication does not hold.

**Proposition 3.3.** *Suppose that the network $N(\mathbf{x})$ Taylor-approximates the homogeneous multivariate polynomial $p(\mathbf{x})$. Then, for every $\epsilon$, there exists a network $N_\epsilon(\mathbf{x})$ that $\epsilon$-approximates $p(\mathbf{x})$, such that $N(\mathbf{x})$ and $N_\epsilon(\mathbf{x})$ have the same number of neurons in each layer. (This statement holds for $\mathbf{x} \in (-R, R)^n$ for any specified $R$.)*

*Proof.* Suppose that $N(\mathbf{x}) = \mathbf{A}_k \sigma(\cdots \sigma(\mathbf{A}_1 \sigma(\mathbf{A}_0 \mathbf{x})) \cdots)$ and that $p(\mathbf{x})$ has degree $d$. Since $p(\mathbf{x})$ is a Taylor approximation of $N(\mathbf{x})$, we can write $N(\mathbf{x})$ as $p(\mathbf{x}) + E(\mathbf{x})$, where $E(\mathbf{x}) = \sum_{i=d+1}^{\infty} E_i(\mathbf{x})$ is a Taylor series with each $E_i(\mathbf{x})$ homogeneous of degree $i$. Since $N(\mathbf{x})$ is the function defined by a neural network, it converges for every $\mathbf{x} \in \mathbb{R}^n$. Thus, $E(\mathbf{x})$ converges, as does $E(\delta \mathbf{x})/\delta^d = \sum_{i=d+1}^{\infty} \delta^{i-d} E_i(\mathbf{x})$. By picking $\delta$ sufficiently small, we can make each term $\delta^{i-d} E_i(\mathbf{x})$ arbitrarily small. Let $\delta$ be small enough that $|E(\delta \mathbf{x})/\delta^d| < \epsilon$ holds for all $\mathbf{x}$ in $(-R, R)^n$.

Let $\mathbf{A}_0' = \delta\mathbf{A}_0$, $\mathbf{A}_k' = \mathbf{A}_k/\delta^d$, and $\mathbf{A}_\ell' = \mathbf{A}_\ell$ for $\ell = 1, 2, \ldots, k - 1$. Then, for $N_\epsilon(\mathbf{x}) = \mathbf{A}_k'\sigma(\cdots\sigma(\mathbf{A}_1'\sigma(\mathbf{A}_0'\mathbf{x}))\cdots)$, we observe that $N_\epsilon(\mathbf{x}) = N(\delta\mathbf{x})/\delta^d$, and therefore:

$$\begin{aligned}
|N_\epsilon(\mathbf{x}) - p(\mathbf{x})| &= |N(\delta\mathbf{x})/\delta^d - p(\mathbf{x})| \\
&= |p(\delta\mathbf{x})/\delta^d + E(\delta\mathbf{x})/\delta^d - p(\mathbf{x})| \\
&= |E(\delta\mathbf{x})/\delta^d| \\
&< \epsilon.
\end{aligned}$$

We conclude that $N_\epsilon(\mathbf{x})$ is an $\epsilon$-approximation of $p(\mathbf{x})$, as desired. $\qquad\square$

For a fixed nonlinear function $\sigma$, we consider the total number of neurons (excluding input and output neurons) needed for a network to approximate a given function. Remarkably, it is possible to attain arbitrarily good approximations of a (not necessarily homogeneous) multivariate polynomial by a feedforward neural network, even with a single hidden layer, without increasing the number of neurons past a certain bound. (See also Corollary 1 in Poggio et al. (2017).)

**Theorem 3.4.** *Suppose that $p(\mathbf{x})$ is a degree-$d$ multivariate polynomial and that the nonlinearity $\sigma$ has nonzero Taylor coefficients up to degree $d$. Let $m_k^\epsilon(p)$ be the minimum number of neurons in a depth-$k$ network that $\epsilon$-approximates $p$. Then, the limit $\lim_{\epsilon\to 0} m_k^\epsilon(p)$ exists (and is finite). (Once again, this statement holds for $\mathbf{x} \in (-R, R)^n$ for any specified $R$.)*

*Proof.* We show that $\lim_{\epsilon\to 0} m_1^\epsilon(p)$ exists; it follows immediately that $\lim_{\epsilon\to 0} m_k^\epsilon(p)$ exists for every $k$, since an $\epsilon$-approximation to $p$ with depth $k$ can be constructed from one with depth 1.

Let $p_1(\mathbf{x}), p_2(\mathbf{x}), \ldots, p_s(\mathbf{x})$ be the monomials of $p(\mathbf{x})$, so that $p(\mathbf{x}) = \sum_i p_i(\mathbf{x})$. We claim that each $p_i(\mathbf{x})$ can be Taylor-approximated by a network $N^i(\mathbf{x})$ with one hidden layer. This follows, for example, from the proof in Lin et al. (2017) that products can be Taylor-approximated by networks with one hidden layer, since each monomial is the product of several inputs (with multiplicity); we prove a far stronger result about $N^i(\mathbf{x})$ later in this paper (see Theorem 4.1).

Suppose now that $N^i(\mathbf{x})$ has $m_i$ hidden neurons. By Proposition 3.3, we conclude that since $p_i(\mathbf{x})$ is homogeneous, it may be $\delta$-approximated by a network $N_\delta^i(\mathbf{x})$ with $m_i$ hidden neurons, where $\delta = \epsilon/s$. By combining the networks $N_\delta^i(\mathbf{x})$ for each $i$, we can define a network $N_\epsilon(\mathbf{x}) = \sum_i N_\delta^i(\mathbf{x})$ with $\sum_i m_i$ neurons. Then, we have:

$$\begin{aligned}
|N_\epsilon(\mathbf{x}) - p(\mathbf{x})| &\leq \sum_i |N_\delta^i(\mathbf{x}) - p_i(\mathbf{x})| \\
&\leq \sum_i \delta = s\delta = \epsilon.
\end{aligned}$$

Hence, $N_\epsilon(\mathbf{x})$ is an $\epsilon$-approximation of $p(\mathbf{x})$, implying that $m_1^\epsilon(p) \leq \sum_i m_i$ for every $\epsilon$. Thus, $\lim_{\epsilon\to 0} m_1^\epsilon(p)$ exists, as desired. $\qquad\square$

This theorem is perhaps surprising, since it is common for $\epsilon$-approximations to functions to require ever-greater complexity, approaching infinity as $\epsilon \to 0$. For example, the function $\exp(|-x|)$ may be approximated on the domain $(-\pi, \pi)$ by Fourier sums of the form $\sum_{k=0}^m a_m \cos(kx)$. However, in order to achieve $\epsilon$-approximation, we need to take $m \sim 1/\sqrt{\epsilon}$ terms. By contrast, we have shown that a finite neural network architecture can achieve arbitrarily good approximations merely by altering its weights.

Note also that the assumption of nonzero Taylor coefficients cannot be dropped from Theorem 3.4. For example, the theorem is false for rectified linear units (ReLUs), which are piecewise linear and do not admit a Taylor series. This is because $\epsilon$-approximating a non-linear polynomial with a piecewise linear function requires an ever-increasing number of pieces as $\epsilon \to 0$.

Theorem 3.4 allows us to make the following definition:

**Definition 3.5.** *Suppose that a nonlinear function $\sigma$ is given. For $p$ a multivariate polynomial, let $m_k^{uniform}(p)$ be the minimum number of neurons in a depth-$k$ network that $\epsilon$-approximates $p$ for*

*all $\epsilon$ arbitrarily small. Set $m^{uniform}(p) = \min_k m_k^{uniform}(p)$. Likewise, let $m_k^{Taylor}(p)$ be the minimum number of neurons in a depth-k network that Taylor-approximates p, and set $m^{Taylor}(p) = \min_k m_k^{Taylor}(p)$.*

In the next section, we will show that there is an exponential gap between $m_1^{\text{uniform}}(p)$ and $m^{\text{uniform}}(p)$ and between $m_1^{\text{Taylor}}(p)$ and $m^{\text{Taylor}}(p)$ for various classes of polynomials $p$.

## 4 THE INEFFICIENCY OF SHALLOW NETWORKS

In this section, we compare the efficiency of shallow networks (those with a single hidden layer) and deep networks at approximating multivariate polynomials. Proofs of our main results are included in the Appendix.

### 4.1 MULTIVARIATE POLYNOMIALS

Our first result shows that uniform approximation of monomials requires exponentially more neurons in a shallow than a deep network.

**Theorem 4.1.** *Let $p(\mathbf{x})$ denote the monomial $x_1^{r_1} x_2^{r_2} \cdots x_n^{r_n}$, with $d = \sum_{i=1}^n r_i$. Suppose that the nonlinearity $\sigma$ has nonzero Taylor coefficients up to degree $2d$. Then, we have:*

(i) $m_1^{uniform}(p) = \prod_{i=1}^n (r_i + 1)$,

(ii) $m^{uniform}(p) \leq \sum_{i=1}^n (7\lceil \log_2(r_i) \rceil + 4)$,

*where $\lceil x \rceil$ denotes the smallest integer that is at least $x$.*

We can prove a comparable result for $m^{\text{Taylor}}$ under slightly weaker assumptions on $\sigma$. Note that by setting $r_1 = r_2 = \ldots = r_n = 1$, we recover the result of Lin et al. (2017) that the product of $n$ numbers requires $2^n$ neurons in a shallow network but can be Taylor-approximated with linearly many neurons in a deep network.

**Theorem 4.2.** *Let $p(\mathbf{x})$ denote the monomial $x_1^{r_1} x_2^{r_2} \cdots x_n^{r_n}$, with $d = \sum_{i=1}^n r_i$. Suppose that $\sigma$ has nonzero Taylor coefficients up to degree $d$. Then, we have:*

(i) $m_1^{Taylor}(p) = \prod_{i=1}^n (r_i + 1)$,

(ii) $m^{Taylor}(p) \leq \sum_{i=1}^n (7\lceil \log_2(r_i) \rceil + 4)$.

It is worth noting that neither of Theorems 4.1 and 4.2 implies the other. This is because it is possible for a polynomial to admit a compact uniform approximation without admitting a compact Taylor approximation.

It is natural now to consider the cost of approximating general polynomials. However, without further constraint, this is relatively uninstructive because polynomials of degree $d$ in $n$ variables live within a space of dimension $\binom{n+d}{d}$, and therefore most require exponentially many neurons for *any* depth of network. We therefore consider polynomials of *sparsity* $c$: that is, those that can be represented as the sum of $c$ monomials. This includes many natural functions.

The following theorem, when combined with Theorems 4.1 and 4.2, shows that general polynomials $p$ with subexponential sparsity have exponentially large $m_1^{\text{uniform}}(p)$ and $m_1^{\text{Taylor}}(p)$, but subexponential $m^{\text{uniform}}(p)$ and $m^{\text{Taylor}}(p)$.

**Theorem 4.3.** *Let $p(\mathbf{x})$ be a multivariate polynomial of degree $d$ and sparsity $c$, having monomials $q_1(\mathbf{x}), q_2(\mathbf{x}), \ldots, q_c(\mathbf{x})$. Suppose that the nonlinearity $\sigma$ has nonzero Taylor coefficients up to degree $2d$. Then, we have:*

(i) $m_1^{uniform}(p) \geq \frac{1}{c} \max_j m_1^{uniform}(q_j)$.

(ii) $m^{uniform}(p) \leq \sum_j m^{uniform}(q_j)$.

*These statements also hold if $m^{uniform}$ is replaced with $m^{Taylor}$.*

As mentioned above with respect to ReLUs, some assumptions on the Taylor coefficients of the activation function are necessary for the results we present. However, it is possible to loosen the assumptions of Theorem 4.1 and 4.2 while still obtaining exponential lower bounds on $m_1^{\text{uniform}}(p)$ and $m_1^{\text{Taylor}}(p)$:

**Theorem 4.4.** *Let $p(\mathbf{x})$ denote the monomial $x_1^{r_1} x_2^{r_2} \cdots x_n^{r_n}$, with $d = \sum_{i=1}^{n} r_i$. Suppose that the nonlinearity $\sigma$ has nonzero $d$th Taylor coefficient (other Taylor coefficients are allowed to be zero). Then, $m_1^{\text{uniform}}(p)$ and $m_1^{\text{Taylor}}(p)$ are at least $\frac{1}{d} \prod_{i=1}^{n} (r_i + 1)$. (An even better lower bound is the maximum coefficient in the polynomial $\prod_i (1 + y + \ldots + y^{r_i})$.)*

## 4.2 UNIVARIATE POLYNOMIALS

As with multivariate polynomials, depth can offer an exponential savings when approximating univariate polynomials. We show below (Proposition 4.5) that a shallow network can approximate any degree-$d$ univariate polynomial with a number of neurons at most linear in $d$. The monomial $x^d$ requires $d + 1$ neurons in a shallow network (Proposition 4.6), but can be approximated with only logarithmically many neurons in a deep network. Thus, depth allows us to reduce networks from linear to logarithmic size, while for multivariate polynomials the gap was between exponential and linear. The difference here arises because the dimensionality of the space of univariate degree-$d$ polynomials is linear in $d$, which the dimensionality of the space of multivariate degree-$d$ polynomials is exponential in $d$.

**Proposition 4.5.** *Suppose that the nonlinearity $\sigma$ has nonzero Taylor coefficients up to degree $d$. Then, $m_1^{\text{Taylor}}(p) \leq d + 1$ for every univariate polynomial $p$ of degree $d$.*

*Proof.* Pick $a_0, a_1, \ldots, a_d$ to be arbitrary, distinct real numbers. Consider the Vandermonde matrix $\mathbf{A}$ with entries $A_{ij} = a_i^j$. It is well-known that $\det(\mathbf{A}) = \prod_{i < i'} (a_{i'} - a_i) \neq 0$. Hence, $\mathbf{A}$ is invertible, which means that multiplying its columns by nonzero values gives another invertible matrix. Suppose that we multiply the $j$th column of $\mathbf{A}$ by $\sigma_j$ to get $\mathbf{A}'$, where $\sigma(x) = \sum_j \sigma_j x^j$ is the Taylor expansion of $\sigma(x)$.

Now, observe that the $i$th row of $\mathbf{A}'$ is exactly the coefficients of $\sigma(a_i x)$, up to the degree-$d$ term. Since $\mathbf{A}'$ is invertible, the rows must be linearly independent, so the polynomials $\sigma(a_i x)$, restricted to terms of degree at most $d$, must themselves be linearly independent. Since the space of degree-$d$ univariate polynomials is $(d + 1)$-dimensional, these $d + 1$ linearly independent polynomials must span the space. Hence, $m_1^{\text{Taylor}}(p) \leq d + 1$ for any univariate degree-$d$ polynomial $p$. In fact, we can fix the weights from the input neuron to the hidden layer (to be $a_0, a_1, \ldots, a_d$, respectively) and still represent any polynomial $p$ with $d + 1$ hidden neurons. $\square$

**Proposition 4.6.** *Let $p(x) = x^d$, and suppose that the nonlinearity $\sigma(x)$ has nonzero Taylor coefficients up to degree $2d$. Then, we have:*

(i) $m_1^{uniform}(p) = d + 1$.

(ii) $m^{uniform}(p) \leq 7 \lceil \log_2(d) \rceil$.

*These statements also hold if $m^{uniform}$ is replaced with $m^{Taylor}$.*

*Proof.* Part (i) follows from part (i) of Theorems 4.1 and 4.2 by setting $n = 1$ and $r_1 = d$.

For part (ii), observe that we can Taylor-approximate the square $x^2$ of an input $x$ with three neurons in a single layer:

$$\frac{1}{2\sigma''(0)} \left( \sigma(x) + \sigma(-x) - 2\sigma(0) \right) = x^2 + \mathcal{O}(x^4 + x^5 + \ldots).$$

We refer to this construction as a *square gate*, and the construction of Lin et al. (2017) as a *product gate*. We also use *identity gate* to refer to a neuron that simply preserves the input of a neuron from the preceding layer (this is equivalent to the *skip connections* in residual nets (He et al., 2016)).

Consider a network in which each layer contains a square gate (3 neurons) and either a product gate or an identity gate (4 or 1 neurons, respectively), according to the following construction: The square gate squares the output of the preceding square gate, yielding inductively a result of the form $x^{2^k}$, where $k$ is the depth of the layer. Writing $d$ in binary, we use a product gate if there is a 1 in the $2^{k-1}$-place; if so, the product gate multiplies the output of the preceding product gate by the output of the preceding square gate. If there is a 0 in the $2^{k-1}$-place, we use an identity gate instead of a product gate. Thus, each layer computes $x^{2^k}$ and multiplies $x^{2^{k-1}}$ to the computation if the $2^{k-1}$-place in $d$ is 1. The process stops when the product gate outputs $x^d$.

This network clearly uses at most $7\lceil \log_2(d) \rceil$ neurons, with a worst case scenario where $d+1$ is a power of 2. Hence $m^{\text{Taylor}}(p) \leq 7\lceil \log_2(d) \rceil$, with $m^{\text{uniform}}(p) \leq m^{\text{Taylor}}(p)$ by Proposition 3.3 since $p$ is homogeneous. □

## 5 HOW EFFICIENCY IMPROVES WITH DEPTH

We now consider how $m_k^{\text{uniform}}(p)$ scales with $k$, interpolating between exponential in $n$ (for $k = 1$) and linear in $n$ (for $k = \log n$). In practice, networks with modest $k > 1$ are effective at representing natural functions. We explain this theoretically by showing that the cost of approximating the product polynomial drops off rapidly as $k$ increases.

By repeated application of the shallow network construction in Lin et al. (2017), we obtain the following upper bound on $m_k^{\text{uniform}}(p)$, which we conjecture to be essentially tight. Our approach leverages the compositionality of polynomials, as discussed e.g. in Mhaskar et al. (2016) and Poggio et al. (2017), using a tree-like neural network architecture.

**Theorem 5.1.** *Let $p(\mathbf{x})$ equal the product $x_1 x_2 \cdots x_n$, and suppose $\sigma$ has nonzero Taylor coefficients up to degree $n$. Then, we have:*

$$m_k^{uniform}(p) = \mathcal{O}\left(n^{(k-1)/k} \cdot 2^{n^{1/k}}\right).$$
(1)

*Proof.* We construct a network in which groups of the $n$ inputs are recursively multiplied up to Taylor approximation. The $n$ inputs are first divided into groups of size $b_1$, and each group is multiplied in the first hidden layer using $2^{b_1}$ neurons (as described in Lin et al. (2017)). Thus, the first hidden layer includes a total of $2^{b_1} n/b_1$ neurons. This gives us $n/b_1$ values to multiply, which are in turn divided into groups of size $b_2$. Each group is multiplied in the second hidden layer using $2^{b_2}$ neurons. Thus, the second hidden layer includes a total of $2^{b_2} n/(b_1 b_2)$ neurons.

We continue in this fashion for $b_1, b_2, \ldots, b_k$ such that $b_1 b_2 \cdots b_k = n$, giving us one neuron which is the product of all of our inputs. By considering the total number of neurons used, we conclude

$$m_k^{\text{Taylor}}(p) \leq \sum_{i=1}^{k} \frac{n}{\prod_{j=1}^{i} b_j} 2^{b_i} = \sum_{i=1}^{k} \left( \prod_{j=i+1}^{k} b_j \right) 2^{b_i}.$$
(2)

By Proposition 3.3, $m_k^{\text{uniform}}(p) \leq m_k^{\text{Taylor}}(p)$ since $p$ is homogeneous. Setting $b_i = n^{1/k}$, for each $i$, gives us the desired bound (1). □

In fact, we can solve for the choice of $b_i$ such that the upper bound in (2) is minimized, under the condition $b_1 b_2 \cdots b_k = n$. Using the technique of Lagrange multipliers, we know that the optimum occurs at a minimum of the function

$$\mathcal{L}(b_i, \lambda) := \left( n - \prod_{i=1}^{k} b_i \right) \lambda + \sum_{i=1}^{k} \left( \prod_{j=i+1}^{k} b_j \right) 2^{b_i}.$$

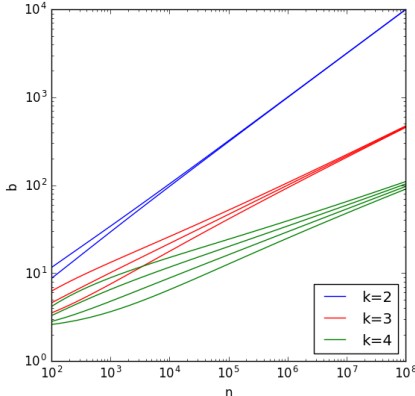

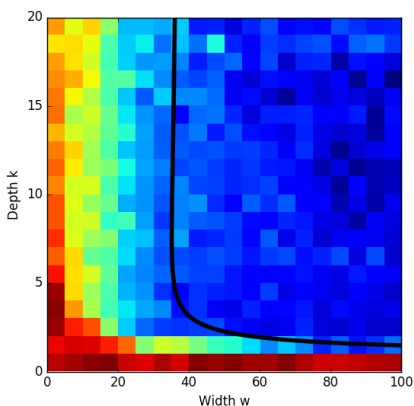

Figure 1: The optimal settings for $\{b_i\}_{i=1}^k$ as $n$ varies are shown for $k = 1, 2, 3$. Observe that the $b_i$ converge to $n^{1/k}$ for large $n$, as witnessed by a linear fit in the log-log plot. The exact values are given by equations (4) and (5).

Figure 2: Performance of trained networks in approximating the product of 20 input variables, ranging from red (high error) to blue (low error). The error shown here is the expected absolute difference between the predicted and actual product. The curve $w = n^{(k-1)/k} \cdot 2^{n^{1/k}}$ for $n = 20$ is shown in black. In the region above and to the right of the curve, it is possible to effectively approximate the product function (Theorem 5.1).

Differentiating $\mathcal{L}$ with respect to $b_i$, we obtain the conditions

$$0 = -\lambda \prod_{\substack{j \neq i}} b_j + \sum_{h=1}^{i-1} \left( \frac{\prod_{j=h+1}^k b_j}{b_i} \right) 2^{b_h} + (\log 2) \left( \prod_{j=i+1}^k b_j \right) 2^{b_i}, \text{ for } 1 \leq i \leq k \qquad (3)$$

$$0 = n - \prod_{j=1}^k b_j. \qquad (4)$$

Dividing (3) by $\prod_{j=i+1}^k b_j$ and rearranging gives us the recursion

$$b_i = b_{i-1} + \log_2(b_{i-1} - 1/\log 2). \qquad (5)$$

Thus, the optimal $b_i$ are not exactly equal but very slowly increasing with $i$ (see Figure 1).

The following conjecture states that the bound given in Theorem 5.1 is (approximately) optimal.

**Conjecture 5.2.** *Let $p(\mathbf{x})$ equal to the product $x_1 x_2 \cdots x_n$, and suppose that $\sigma$ has all nonzero Taylor coefficients. Then, we have:*

$$m_k^{uniform}(p) = 2^{\Theta(n^{1/k})}, \qquad (6)$$

i.e., the exponent grows as $n^{1/k}$ for $n \to \infty$.

We empirically tested Conjecture 5.2 by training ANNs to predict the product of input values $x_1, \ldots, x_n$ with $n = 20$ (see Figure 2). The rapid interpolation from exponential to linear width aligns with our predictions.

In our experiments, we used feedforward networks with dense connections between successive layers. In the figure, we show results for $\sigma(x) = \tanh(x)$ (note that this behavior is even better than expected, since this function actually has numerous zero Taylor coefficients). Similar results were also obtained for rectified linear units (ReLUs) as the nonlinearity, despite the fact that this function

does not even admit a Taylor series. The number of layers was varied, as was the number of neurons within a single layer. The networks were trained using the AdaDelta optimizer (Zeiler, 2012) to minimize the absolute value of the difference between the predicted and actual values. Input variables $x_i$ were drawn uniformly at random from the interval $[0, 2]$, so that the expected value of the output would be of manageable size.

Eq. (6) provides a helpful rule of thumb for how deep is deep enough. Suppose, for instance, that we wish to keep typical layers no wider than about a thousand ($\sim 2^{10}$) neurons. Eq. (6) then implies $n^{1/k} \lesssim 10$, *i.e.*, that the number of layers should be at least

$$k \gtrsim \log_{10} n.$$

It would be very interesting if one could show that general polynomials $p$ in $n$ variables require a superpolynomial number of neurons to approximate for any constant number of hidden layers. The analogous statement for Boolean circuits - whether the complexity classes $TC^0$ and $TC^1$ are equal - remains unresolved and is assumed to be quite hard. Note that the formulations for Boolean circuits and deep neural networks are independent statements (neither would imply the other) due to the differences between computation on binary and real values. Indeed, gaps in expressivity have already been proven to exist for real-valued neural networks of different depths, for which the analogous results remain unknown in Boolean circuits (see e.g. Mhaskar (1993); Chui et al. (1994; 1996); Montufar et al. (2014); Cohen et al. (2016); Telgarsky (2016)).

# 6 CONCLUSION

We have shown how the power of deeper ANNs can be quantified even for simple polynomials. We have proved that arbitrarily good approximations of polynomials are possible even with a fixed number of neurons and that there is an exponential gap between the width of shallow and deep networks required for approximating a given sparse polynomial. For $n$ variables, a shallow network requires size exponential in $n$, while a deep network requires at most linearly many neurons. Networks with a constant number $k > 1$ of hidden layers appear to interpolate between these extremes, following a curve exponential in $n^{1/k}$. This suggests a rough heuristic for the number of layers required for approximating simple functions with neural networks. For example, if we want no layers to have more than $2^{10}$ neurons, say, then the minimum number of layers required grows only as $\log_{10} n$. To further improve efficiency using the $\mathcal{O}(n)$ constructions we have presented, it suffices to increase the number of layers by a factor of $\log_2 10 \approx 3$, to $\log_2 n$.

The key property we use in our constructions is compositionality, as detailed in Poggio et al. (2017). It is worth noting that as a consequence our networks enjoy the property of *locality* mentioned in Cohen et al. (2016), which is also a feature of convolutional neural nets. That is, each neuron in a layer is assumed to be connected only to a small subset of neurons from the previous layer, rather than the entirety (or some large fraction). In fact, we showed (e.g. Prop. 4.6) that there exist natural functions computable with linearly many neurons, with each neuron is connected to at most *two* neurons in the preceding layer, which nonetheless cannot be computed with fewer than exponentially many neurons in a single layer, no matter how may connections are used. Our construction can also be framed with reference to the other properties mentioned in Cohen et al. (2016): those of *sharing* (in which weights are shared between neural connections) and *pooling* (in which layers are gradually collapsed, as our construction essentially does with recursive combination of inputs).

This paper has focused exclusively on the resources (neurons and synapses) required to *compute* a given function for fixed network depth. (Note also results of Lu et al. (2017); Hanin & Sellke (2017); Hanin (2017) for networks of fixed width.) An important complementary challenge is to quantify the resources (e.g. training steps) required to *learn* the computation, i.e., to converge to appropriate weights using training data — possibly a fixed amount thereof, as suggested in Zhang et al. (2017). There are simple functions that can be computed with polynomial resources but require exponential resources to learn (Shalev-Shwartz et al., 2017). It is quite possible that architectures we have not considered increase the feasibility of learning. For example, residual networks (ResNets) (He et al., 2016) and unitary nets (see e.g. Arjovsky et al. (2016); Jing et al. (2017)) are no more powerful in representational ability than conventional networks of the same size, but by being less susceptible to the "vanishing/exploding gradient" problem, it is far easier to optimize them in practice. We look forward to future work that will help us understand the power of neural networks to learn.

## 7  ACKNOWLEDGMENTS

This work was supported by the Foundational Questions Institute http://fqxi.org/, the Rothberg Family Fund for Cognitive Science and NSF grant 1122374. We would like to thank Tomaso Poggio, Scott Aaronson, Surya Ganguli, David Budden, Henry Lin, and the anonymous referees for helpful suggestions and discussions, and the Center for Brains, Minds, & Machines for an excellent working environment.

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

## APPENDIX

### 7.1 PROOF OF THEOREM 4.1.

Without loss of generality, suppose that $r_i > 0$ for $i = 1, \ldots, n$. Let $X$ be the multiset in which $x_i$ occurs with multiplicity $r_i$.

We first show that $\prod_{i=1}^{n}(r_i + 1)$ neurons are *sufficient* to approximate $p(\mathbf{x})$. Appendix A in Lin et al. (2017) demonstrates that for variables $y_1, \ldots, y_N$, the product $y_1 \cdot \cdots \cdot y_N$ can be Taylor-approximated as a linear combination of the $2^N$ functions $\sigma(\pm y_1 \pm \cdots \pm y_d)$.

Consider setting $y_1, \ldots, y_d$ equal to the elements of multiset $X$. Then, we conclude that we can approximate $p(\mathbf{x})$ as a linear combination of the functions $\sigma(\pm y_1 \pm \cdots \pm y_d)$. However, these functions are not all distinct: there are $r_i + 1$ distinct ways to assign $\pm$ signs to $r_i$ copies of $x_i$ (ignoring permutations of the signs). Therefore, there are $\prod_{i=1}^{n}(r_i + 1)$ distinct functions $\sigma(\pm y_1 \pm$

$\cdots \pm y_N$), proving that $m^{\text{Taylor}}(p) \leq \prod_{i=1}^{n}(r_i + 1)$. Proposition 3.3 implies that for homogeneous polynomials $p$, we have $m_1^{\text{uniform}}(p) \leq m_1^{\text{Taylor}}(p)$.

We now show that this number of neurons is also *necessary* for approximating $p(\mathbf{x})$. Suppose that $N_\epsilon(\mathbf{x})$ is an $\epsilon$-approximation to $p(\mathbf{x})$ with depth 1, and let the Taylor series of $N_\epsilon(\mathbf{x})$ be $p(\mathbf{x}) + E(\mathbf{x})$. Let $E_k(\mathbf{x})$ be the degree-$k$ homogeneous component of $E(\mathbf{x})$, for $0 \leq k \leq 2d$. By the definition of $\epsilon$-approximation, $\sup_{\mathbf{x}} E(\mathbf{x})$ goes to 0 as $\epsilon$ does, so by picking $\epsilon$ small enough, we can ensure that the coefficients of each $E_k(\mathbf{x})$ go to 0.

Let $m = m_1^{\text{uniform}}(p)$ and suppose that $\sigma(x)$ has the Taylor expansion $\sum_{k=0}^{\infty} \sigma_k x^k$. Then, by grouping terms of each order, we conclude that there exist constants $a_{ij}$ and $w_j$ such that

$$\sigma_d \sum_{j=1}^{m} w_j \left( \sum_{i=1}^{n} a_{ij} x_i \right)^d = p(\mathbf{x}) + E_d(\mathbf{x})$$

$$\sigma_k \sum_{j=1}^{m} w_j \left( \sum_{i=1}^{n} a_{ij} x_i \right)^k = E_k(\mathbf{x}) \quad \text{for } k \neq d.$$

For each $S \subseteq X$, let us take the derivative of this equation by every variable that occurs in $S$, where we take multiple derivatives of variables that occur multiple times. This gives

$$\frac{\sigma_d \cdot d!}{|S|!} \sum_{j=1}^{m} w_j \prod_{h \in S} a_{hj} \left( \sum_{i=1}^{n} a_{ij} x_i \right)^{d-|S|} = \frac{\partial}{\partial S} p(\mathbf{x}) + \frac{\partial}{\partial S} E_d(\mathbf{x}), \tag{7}$$

$$\frac{\sigma_k \cdot k!}{|S|!} \sum_{j=1}^{m} w_j \prod_{h \in S} a_{hj} \left( \sum_{i=1}^{n} a_{ij} x_i \right)^{k-|S|} = \frac{\partial}{\partial S} E_k(\mathbf{x}). \tag{8}$$

Observe that there are $r \equiv \prod_{i=1}^{n}(r_i + 1)$ choices for $S$, since each variable $x_i$ can be included anywhere from 0 to $r_i$ times. Define $\mathbf{A}$ to be the $r \times m$ matrix with entries $A_{S,j} = \prod_{h \in S} a_{hj}$. We claim that $\mathbf{A}$ has full row rank. This would show that the number of columns $m$ is at least the number of rows $r = \prod_{i=1}^{n}(r_i + 1)$, proving the desired lower bound on $m$.

Suppose towards contradiction that the rows $A_{S_\ell, \bullet}$ admit a linear dependence:

$$\sum_{\ell=1}^{r} c_\ell A_{S_\ell, \bullet} = \mathbf{0},$$

where the coefficients $c_\ell$ are all nonzero and the $S_\ell$ denote distinct subsets of $X$. Let $S_*$ be such that $|c_*|$ is maximized. Then, take the dot product of each side of the above equation by the vector with entries (indexed by $j$) equal to $w_j \left( \sum_{i=1}^{n} a_{ij} x_i \right)^{d-|S_*|}$:

$$0 = \sum_{\ell=1}^{r} c_\ell \sum_{j=1}^{m} w_j \prod_{h \in S_\ell} a_{hj} \left( \sum_{i=1}^{n} a_{ij} x_i \right)^{d-|S_*|}$$

$$= \sum_{\ell \mid (|S_\ell| = |S_*|)} c_\ell \sum_{j=1}^{m} w_j \prod_{h \in S_\ell} a_{hj} \left( \sum_{i=1}^{n} a_{ij} x_i \right)^{d-|S_\ell|}$$

$$+ \sum_{\ell \mid (|S_\ell| \neq |S_*|)} c_\ell \sum_{j=1}^{m} w_j \prod_{h \in S_\ell} a_{hj} \left( \sum_{i=1}^{n} a_{ij} x_i \right)^{(d+|S_\ell|-|S_*|)-|S_\ell|}.$$

We can use (7) to simplify the first term and (8) (with $k = d + |S_\ell| - |S_*|$) to simplify the second term, giving us:

$$0 = \sum_{\ell \mid (|S_\ell| = |S_*|)} c_\ell \cdot \frac{|S_\ell|!}{\sigma_d \cdot d!} \cdot \left( \frac{\partial}{\partial S_\ell} p(\mathbf{x}) + \frac{\partial}{\partial S_\ell} E_d(\mathbf{x}) \right) \tag{9}$$

$$+ \sum_{\ell \mid (|S_\ell| \neq |S_*|)} c_\ell \cdot \frac{|S_\ell|!}{\sigma_{d+|S_\ell|-|S_*|} \cdot (d + |S_\ell| - |S_*|)!} \cdot \frac{\partial}{\partial S_\ell} E_{d+|S_\ell|-|S_*|}(\mathbf{x})$$

Consider the coefficient of the monomial $\frac{\partial}{\partial S_*} p(\mathbf{x})$, which appears in the first summand with coefficient $c_* \cdot \frac{|S_*|!}{\sigma_d \cdot d!}$. Since the $S_\ell$ are distinct, this monomial does not appear in any other term $\frac{\partial}{\partial S_\ell} p(\mathbf{x})$, but it could appear in some of the terms $\frac{\partial}{\partial S_\ell} E_k(\mathbf{x})$.

By definition, $|c_*|$ is the largest of the values $|c_\ell|$, and by setting $\epsilon$ small enough, all coefficients of $\frac{\partial}{\partial S_\ell} E_k(\mathbf{x})$ can be made negligibly small for every $k$. This implies that the coefficient of the monomial $\frac{\partial}{\partial S_*} p(\mathbf{x})$ can be made arbitrarily close to $c_* \cdot \frac{|S_*|!}{\sigma_d \cdot d!}$, which is nonzero since $c_*$ is nonzero.

However, the left-hand side of equation (9) tells us that this coefficient should be zero - a contradiction. We conclude that $\mathbf{A}$ has full row rank, and therefore that $m_1^{\text{uniform}}(p) = m \geq \prod_{i=1}^{n}(r_i + 1)$. This completes the proof of part (i).

We now consider part (ii) of the theorem. It follows from Proposition 4.6, part (ii) that, for each $i$, we can Taylor-approximate $x_i^{r_i}$ using $7\lceil \log_2(r_i) \rceil$ neurons arranged in a deep network. Therefore, we can Taylor-approximate all of the $x_i^{r_i}$ using a total of $\sum_i 7\lceil \log_2(r_i) \rceil$ neurons. From Lin et al. (2017), we know that these $n$ terms can be multiplied using $4n$ additional neurons, giving us a total of $\sum_i (7\lceil \log_2(r_i) \rceil + 4)$. Proposition 3.3 implies again that $m_1^{\text{uniform}}(p) \leq m_1^{\text{Taylor}}(p)$. This completes the proof.

## 7.2 Proof of Theorem 4.2.

As above, suppose that $r_i > 0$ for $i = 1, \ldots, n$, and let $X$ be the multiset in which $x_i$ occurs with multiplicity $r_i$.

It is shown in the proof of Theorem 4.1 that $\prod_{i=1}^{n}(r_i + 1)$ neurons are *sufficient* to Taylor-approximate $p(x)$. We now show that this number of neurons is also *necessary* for approximating $p(\mathbf{x})$. Let $m = m_1^{\text{Taylor}}(p)$ and suppose that $\sigma(x)$ has the Taylor expansion $\sum_{k=0}^{\infty} \sigma_k x^k$. Then, by grouping terms of each order, we conclude that there exist constants $a_{ij}$ and $w_j$ such that

$$\sigma_d \sum_{j=1}^{m} w_j \left( \sum_{i=1}^{n} a_{ij} x_i \right)^d = p(\mathbf{x}) \tag{10}$$

$$\sigma_k \sum_{j=1}^{m} w_j \left( \sum_{i=1}^{n} a_{ij} x_i \right)^k = 0 \quad \text{for } 0 \leq k \leq N - 1. \tag{11}$$

For each $S \subseteq X$, let us take the derivative of equations (10) and (11) by every variable that occurs in $S$, where we take multiple derivatives of variables that occur multiple times. This gives

$$\frac{\sigma_d \cdot d!}{|S|!} \sum_{j=1}^{m} w_j \prod_{h \in S} a_{hj} \left( \sum_{i=1}^{n} a_{ij} x_i \right)^{d - |S|} = \frac{\partial}{\partial S} p(\mathbf{x}), \tag{12}$$

$$\frac{\sigma_k \cdot k!}{|S|!} \sum_{j=1}^{m} w_j \prod_{h \in S} a_{hj} \left( \sum_{i=1}^{n} a_{ij} x_i \right)^{k - |S|} = 0 \tag{13}$$

for $|S| \leq k \leq d - 1$. Observe that there are $r = \prod_{i=1}^{n}(r_i + 1)$ choices for $S$, since each variable $x_i$ can be included anywhere from 0 to $r_i$ times. Define $\mathbf{A}$ to be the $r \times m$ matrix with entries $A_{S,j} = \prod_{h \in S} a_{hj}$. We claim that $\mathbf{A}$ has full row rank. This would show that the number of columns $m$ is at least the number of rows $r = \prod_{i=1}^{n}(r_i + 1)$, proving the desired lower bound on $m$.

Suppose towards contradiction that the rows $A_{S_\ell, \bullet}$ admit a linear dependence:

$$\sum_{\ell=1}^{r} c_\ell A_{S_\ell, \bullet} = \mathbf{0},$$

where the coefficients $c_\ell$ are nonzero and the $S_\ell$ denote distinct subsets of $X$. Set $s = \max_\ell |S_\ell|$. Then, take the dot product of each side of the above equation by the vector with entries (indexed by

$j$) equal to $w_j \left( \sum_{i=1}^{n} a_{ij} x_i \right)^{d-s}$:

$$
\begin{aligned}
0 &= \sum_{\ell=1}^{r} c_\ell \sum_{j=1}^{m} w_j \prod_{h \in S_\ell} a_{hj} \left( \sum_{i=1}^{n} a_{ij} x_i \right)^{d-s} \\
&= \sum_{\ell \mid (|S_\ell|=s)} c_\ell \sum_{j=1}^{m} w_j \prod_{h \in S_\ell} a_{hj} \left( \sum_{i=1}^{n} a_{ij} x_i \right)^{d-|S_\ell|} \\
&+ \sum_{\ell \mid (|S_\ell|<s)} c_\ell \sum_{j=1}^{m} w_j \prod_{h \in S_\ell} a_{hj} \left( \sum_{i=1}^{n} a_{ij} x_i \right)^{(d+|S_\ell|-s)-|S_\ell|} .
\end{aligned}
$$

We can use (12) to simplify the first term and (13) (with $k = d + |S_\ell| - s$) to simplify the second term, giving us:

$$
\begin{aligned}
0 &= \sum_{\ell \mid (|S_\ell|=s)} c_\ell \cdot \frac{|S_\ell|!}{\sigma_d \cdot d!} \cdot \frac{\partial}{\partial S_\ell} p(\mathbf{x}) + \sum_{\ell \mid (|S_\ell|<s)} c_\ell \cdot \frac{|S_\ell|!}{\sigma_{d+|S_\ell|-s} \cdot (d+|S_\ell|-s)!} \cdot 0 \\
&= \sum_{\ell \mid (|S_\ell|=s)} c_\ell \cdot \frac{|S_\ell|!}{\sigma_d \cdot d!} \cdot \frac{\partial}{\partial S_\ell} p(\mathbf{x}).
\end{aligned}
$$

Since the distinct monomials $\frac{\partial}{\partial S_\ell} p(\mathbf{x})$ are linearly independent, this contradicts our assumption that the $c_\ell$ are nonzero. We conclude that $\mathbf{A}$ has full row rank, and therefore that $m_1^{\text{Taylor}}(p) = m \geq \prod_{i=1}^{n}(r_i + 1)$. This completes the proof of part (i).

Part (ii) of the theorem was demonstrated in the proof of Theorem 4.1. This completes the proof.

### 7.3    PROOF OF THEOREM 4.3.

Our proof in Theorem 4.1 relied upon the fact that all nonzero partial derivatives of a monomial are linearly independent. This fact is not true for general polynomials $p$; however, an exactly similar argument shows that $m_1^{\text{uniform}}(p)$ is at least the number of linearly independent partial derivatives of $p$, taken with respect to multisets of the input variables.

Consider the monomial $q$ of $p$ such that $m_1^{\text{uniform}}(q)$ is maximized, and suppose that $q(\mathbf{x}) = x_1^{r_1} x_2^{r_2} \cdots x_n^{r_n}$. By Theorem 4.1, $m_1^{\text{uniform}}(q)$ is equal to the number $\prod_{i=1}^{n}(r_i + 1)$ of distinct monomials that can be obtained by taking partial derivatives of $q$. Let $Q$ be the set of such monomials, and let $D$ be the set of (iterated) partial derivatives corresponding to them, so that for $d \in D$, we have $d(q) \in Q$.

Consider the set of polynomials $P = \{d(p) \mid d \in D\}$. We claim that there exists a linearly independent subset of $P$ with size at least $|D|/c$. Suppose to the contrary that $P'$ is a maximal linearly independent subset of $P$ with $|P'| < |D|/c$.

Since $p$ has $c$ monomials, every element of $P$ has at most $c$ monomials. Therefore, the total number of distinct monomials in elements of $P'$ is less than $|D|$. However, there are at least $|D|$ distinct monomials contained in elements of $P$, since for $d \in D$, the polynomial $d(p)$ contains the monomial $d(q)$, and by definition all $d(q)$ are distinct as $d$ varies. We conclude that there is some polynomial $p' \in P \backslash P'$ containing a monomial that does not appear in any element of $P'$. But then $p'$ is linearly independent of $P'$, a contradiction since we assumed that $P'$ was maximal.

We conclude that some linearly independent subset of $P$ has size at least $|D|/c$, and therefore that the space of partial derivatives of $p$ has rank at least $|D|/c = m_1^{\text{uniform}}(q)/c$. This proves part (i) of the theorem. Part (ii) follows immediately from the definition of $m^{\text{uniform}}(p)$.

Similar logic holds for $m^{\text{Taylor}}$.

## 7.4 PROOF OF THEOREM 4.4.

We will prove the desired lower bounds for $m_1^{\text{uniform}}(p)$; a very similar argument holds for $m_1^{\text{Taylor}}(p)$. As above, suppose that $r_i > 0$ for $i = 1, \ldots, n$. Let $X$ be the multiset in which $x_i$ occurs with multiplicity $r_i$.

Suppose that $N_\epsilon(\mathbf{x})$ is an $\epsilon$-approximation to $p(\mathbf{x})$ with depth 1, and let the degree-$d$ Taylor polynomial of $N_\epsilon(\mathbf{x})$ be $p(\mathbf{x}) + E(\mathbf{x})$. Let $E_d(\mathbf{x})$ be the degree-$d$ homogeneous component of $E(\mathbf{x})$. Observe that the coefficients of the error polynomial $E_d(\mathbf{x})$ can be made arbitrarily small by setting $\epsilon$ sufficiently small.

Let $m = m_1^{\text{uniform}}(p)$ and suppose that $\sigma(x)$ has the Taylor expansion $\sum_{k=0}^{\infty} \sigma_k x^k$. Then, by grouping terms of each order, we conclude that there exist constants $a_{ij}$ and $w_j$ such that

$$\sigma_d \sum_{j=1}^{m} w_j \left( \sum_{i=1}^{n} a_{ij} x_i \right)^d = p(\mathbf{x}) + E_d(\mathbf{x})$$

For each $S \subseteq X$, let us take the derivative of this equation by every variable that occurs in $S$, where we take multiple derivatives of variables that occur multiple times. This gives

$$\frac{\sigma_d \cdot d!}{|S|!} \sum_{j=1}^{m} w_j \prod_{h \in S} a_{hj} \left( \sum_{i=1}^{n} a_{ij} x_i \right)^{d-|S|} = \frac{\partial}{\partial S} p(\mathbf{x}) + \frac{\partial}{\partial S} E_d(\mathbf{x}).$$

Consider this equation as $S \subseteq X$ varies over all $C_s$ multisets of fixed size $s$. The left-hand side represents a linear combination of the $m$ terms $\left( \sum_{i=1}^{n} a_{ij} x_i \right)^{d-s}$. The polynomials $\frac{\partial}{\partial S} p(\mathbf{x}) + \frac{\partial}{\partial S} E_d(\mathbf{x})$ on the right-hand side must be linearly independent as $S$ varies, since the distinct monomials $\frac{\partial}{\partial S} p(\mathbf{x})$ are linearly independent and the coefficients of $\frac{\partial}{\partial S} E_d(\mathbf{x})$ can be made arbitrarily small.

This means that the number $m$ of linearly combined terms on the left-hand side must be at least the number $C_s$ of choices for $S$. Observe that $C_s$ is the coefficient of the term $y^s$ in the polynomial $g(y) = \prod_i (1 + y + \ldots + y^{r_i})$. A simple (and not very good) lower bound for $C_s$ is $\frac{1}{d} \prod_{i=1}^{n} (r_i + 1)$, since there are $\prod_{i=1}^{n} (r_i + 1)$ distinct sub-multisets of $X$, and their cardinalities range from 0 to $d$.

