# OpenReview forum: "The power of deeper networks for expressing natural functions"
_ICLR.cc/2018/Conference — Accept (Poster)_

### Official Review · AnonReviewer3 · 2017-11-27
**a main theorem on polynomial approximation of deep vs shallow neural networks**

**Rating:** 7
**Confidence:** 4

**Review:**

Experimental results have shown that deep networks (many hidden layers) can approximate more complicated functions with less neurons compared to shallow (single hidden layer) networks.
This paper gives an explicit proof when the function in question is a sparse polynomial, ie: a polynomial in n variables, which equals a sum J of monomials of degree at most c.
In this setup, Theorem 4.3 says that a shallow network need at least ~ (1 + c/n)^n many neurons, while the optimal deep network (whose depth is optimized to approximate this particular input polynomial) needs at most  ~ J*n, that is, linear in the number of terms and the number of variables. The paper also has bounds for neural networks of a specified depth k (Theorem 5.1), and the authors conjecture this bound to be tight (Conjecture 5.2).

This is an interesting result, and is an improvement over Lin 2017 (where a similar bound is presented for monomial approximation).
Overall, I like the paper.

Pros: new and interesting result, theoretically sound.
Cons: nothing major.
Comments and clarifications:
* What about the ability of a single neural network to approximate a class of functions (instead of a single p), where the topology is fixed but the network weights are allowed to vary? Could you comment on this problem?
* Is the assumption that \sigma has Taylor expansion to order d tight? (That is, are there counter examples for relaxations of this assumption?)
* As noted, the assumptions of your theorems 4.1-4.3 do not apply to ReLUs, but ReLUs network perform well in practice. Could you provide some further comments on this?

---

> ### Author Response · Authors · 2018-01-05
> **Response to review**
>
> Thank you for this thoughtful feedback. To respond to the particular comments raised:
>
> - This is a very interesting question. In this work, we have supposed that connections between layers of a network are dense. In this case, the topology is given simply by the number of neurons in each layer, and this architecture is relatively versatile. Architectures of the form described in the proof of Thm. 5.1 (where the sizes of the hidden layers follow a decreasing geometric progression) should be especially flexible, able to learn a wide range of monomials and sums of monomials. Intuitively, this network architecture learns well because the initial large hidden layers capture many lower order correlations between input variables, which are then used to calculate higher-order correlations deeper within the network.
>
> - The conditions on the activation function appear to be at least largely tight. As we mention in the text, Thm. 3.4 fails for ReLU activation (where the Taylor series is not even defined), implying that all subsequent theorems also fail for ReLUs. More interestingly, it is possible to multiply d inputs with (slightly) fewer than 2^d neurons if the constant term in the Taylor series for the activation function is zero. We had previously proven that a less elegant exponential bound still holds as long as the dth Taylor coefficient itself is nonzero (without any assumptions on the other coefficients), and we have included this in our revision.
>
> - In practice, we are rarely concerned with uniform approximation for epsilon truly arbitrarily small. ReLUs can be (imperfectly) approximated by Taylor-approximable functions, and the behavior diverges as the desired epsilon decreases. In running our experiments, we observed similar behavior with ReLUs as with Taylor-approximable activation functions, even though the full power of our theoretical results is indeed not applicable.

---

### Official Review · AnonReviewer1 · 2017-11-27
**This paper explores the representation of polynomials up to a given maximum degree by deep networks, demonstrating gaps between deep and shallow architectures.**

**Rating:** 6
**Confidence:** 4

**Review:**

The paper investigates the representation of polynomials by neural networks up to a certain degree and implied uniform approximations. It shows exponential gaps between the width of shallow and deep networks required for approximating a given sparse polynomial.

By focusing on polynomials, the paper is able to use of a variety of tools (e.g. linear algebra) to investigate the representation question. Results such as Proposition 3.3 relate the representation of a polynomial up to a certain degree, to the approximation question. Here it would be good to be more specific about the domain, however, as approximating the low order terms certainly does not guarantee a global uniform approximation.

Theorem 3.4 makes an interesting claim, that a finite network size is sufficient to achieve the best possible approximation of a polynomial (the proof building on previous results, e.g. by Lin et al that I did not verify). The idea being to construct a superposition of Taylor approximations of the individual monomials. Here it would be good to be more specific about the domain. Also, in the discussion of Taylor series, it would be good to mention the point around which the series is developed, e.g. the origin.

The paper mentions that ``the theorem is false for rectified linear units (ReLUs), which are piecewise linear and do not admit a Taylor series''. However, a ReLU can also be approximated by a smooth function and a Taylor series.

Theorem 4.1 seems to be implied by Theorem 4.2. Similarly, parts of Section 4.2 seem to follow directly from the previous discussion.

In page 1 ```existence proofs' without explicit constructions'' This is not true, with numerous papers providing explicit constructions of functions that are representable by neural networks with specific types of activation functions.

---

> ### Author Response · Authors · 2018-01-05
> **Response to review**
>
> We are very grateful for this helpful feedback, and have responded below to individual issues raised.
>
> Thank you for the suggestion that we make clearer the domain under which Prop. 3.3 and Thm. 3.4 hold. We have made explicit in our revision that these results hold for any (fixed) domain (-R, R)^n, and that Taylor series are constructed around the origin.
>
> While it is indeed true that a ReLU can be approximated by a smooth function with a well-defined Taylor series, any particular choice of such a function would fail our strict requirement of uniform approximation for arbitrarily small \epsilon. Since we have assumed that the choice of nonlinear function \sigma is fixed, we cannot use progressively better approximations to ReLUs. Another way of thinking about this is to note that a neural network with ReLUs is ultimately piecewise linear. For a fixed budget of neurons, the number of linear pieces is bounded. Given a fixed number of linear pieces and a general polynomial to approximate, the approximation cannot be better than some fixed \epsilon (depending on the polynomial), whereas we would like \epsilon to be arbitrarily small.
>
> Theorems 4.1 and 4.2 are in fact independent, with neither implying the other. This is because it is possible for a polynomial to admit a compact uniform approximation without admitting a compact Taylor approximation. We have made this clearer in the text.
>
> We have rephrased our discussion of prior literature to emphasize that “existence proofs” are a feature only of *some* of the prior work. There are indeed excellent papers that provide explicit constructions.

---

### Official Review · AnonReviewer2 · 2017-11-27
**Proves exponential improvement for expressing polynomial functions using deep NNs, generalizes Lin et al.**

**Rating:** 6
**Confidence:** 4

**Review:**

Summary and significance: The authors prove that for expressing simple multivariate monomials over n variables, networks of depth 1 require exp(n) many neurons, whereas networks of depth n can represent these monomials using only O(n) neurons.
The paper provides a simple and clear explanation for the important problem of theoretically explaining the power of deep networks, and quantifying the improvement provided by depth.

+ves:
Explaining the power of depth in NNs is fundamental to an understanding of deep learning. The paper is very easy to follow. and the proofs are clearly written. The theorems provide exponential gaps for very simple polynomial functions.

-ves:
1. My main concern with the paper is the novelty of the contribution to the techniques. The results in the paper are more general than that of Lin et al., but the proofs are basically the same, and it's difficult to see the contribution of this paper in terms of the contributing fundamentally new ideas.
2. The second concern is that the results apply only to non-linear activation functions with sufficiently many non-zero derivatives (same requirements as for the results of Lin et al.).
3. Finally, in prop 3.3, reducing from uniform approximations to Taylor approximations, the inequality |E(δx)| <= δ^(d+1) |N(x) - p(x)| does not follow from the definition of a Taylor approximation.

Despite these criticisms, I contend that the significance of the problem, and the clean and understandable results in the paper make it a decent paper for ICLR.

---

> ### Author Response · Authors · 2018-01-05
> **Response to review**
>
> We are very grateful for this close reading and constructive comments. Detailed responses follow:
>
> 1. We believe that in addition to presenting more general results than in the literature, we also contribute techniques that are significantly stronger than those in Lin et al. In particular, tighter proof techniques are required in order to prove lower bounds on the number of neurons required for a uniform approximation. One of the more interesting methodological insights resulting from our approach is that even though uniform approximation does not imply Taylor approximation, we can still use the lack of a Taylor approximation as a significant step towards proving the lack of a uniform approximation. To the best of our knowledge, this is the first time that Taylor approximation and uniform approximation of neural networks have been rigorously linked.
>
> 2. The assumptions on the activation function can be weakened somewhat at the expense of less elegant formulations. We had previously proven that an exponential bound still holds as long as the dth Taylor coefficient itself is nonzero (without any assumptions on the other coefficients), and we have included this statement and proof in our revision. As we mention in the text, Thm. 3.4 fails for ReLU activation (where the Taylor series is not even defined), implying that all subsequent theorems also fail for ReLUs. In practice, however, we are rarely concerned with uniform approximation for epsilon truly arbitrarily small. ReLUs can be (imperfectly) approximated by Taylor-approximable functions, and the behavior diverges as the desired epsilon decreases. In running our experiments, we observed similar behavior with ReLUs as with Taylor-approximable activation functions, even though the full power of our theoretical results is indeed not applicable.
>
> 3. In our revision, we have rewritten the proof of prop. 3.3 to encompass all cases. Thank you for calling this to our attention.

---

### Decision · Program_Chairs · 2018-01-29
**ICLR 2018 Conference Acceptance Decision**

**Decision:**

Accept (Poster)

**Comment:**

All the reviewers are agree on the significance of the topic of understanding expressivity of deep networks. This paper makes good progress in analyzing the ability of deep networks to fit multivariate polynomials. They show exponential depth advantage for general sparse polynomials.

 I am very surprised that the paper misses the original contribution of Andrew Barron. He analyzes the size of the shallow neural networks needed to fit a wide class of functions including polynomials. The deep learning community likes to think that everything has been invented in the current decade.

@article{barron1994approximation,
  title={Approximation and estimation bounds for artificial neural networks},
  author={Barron, Andrew R},
  journal={Machine Learning},
  volume={14},
  number={1},
  pages={115--133},
  year={1994},
  publisher={Springer}
}